# Establishment of a salt-induced bioremediation platform from marine *Vibrio natriegens*

Ling Huang[1], Jun Ni[1✉], Chao Zhong [2], Ping Xu [1], Junbiao Dai [2] & Hongzhi Tang [1✉]

Industrial wastewater discharge, agricultural production, marine shipping, oil extraction, and other activities have caused serious marine pollution, including microplastics, petroleum and its products, heavy metals, pesticides, and other organics. Efficiency of bioremediation of marine pollutions may be limited by high salt concentrations (>1%, w/v), which can cause an apparent loss of microbial activities. In this study, functional promoters P1, P2-1, and P2-2 censoring salt stress were isolated and identified from a *Vibrio natriegens* strain Vmax. Three salt-induced degradation models were constructed to degrade polyethylene terephthalate (PET), chlorpyrifos (CP), and hexabromocyclododecanes (HBCDs) using the marine strain Vmax. The engineered strains are efficient for degradation of the corresponding substrates, with the degradation rates at 15 mg/L PET in 8 d, 50 mg/L CP in 24 h, and 1 mg/L HBCDs in 4 h, respectively. In addition, an immobilization strategy for recycling and reusing of engineered strains was realized by expressing the chitin-binding protein GbpA. This study may help answer the usage of rapidly growing marine bacteria such as *V. natriegens* Vmax to degrade marine pollution efficiently.

[1] State Key Laboratory of Microbial Metabolism, and School of Life Sciences & Biotechnology, Shanghai Jiao Tong University, Shanghai, PR China. [2] CAS Key Laboratory of Quantitative Engineering Biology, Guangdong Provincial Key Laboratory of Synthetic Genomics and Shenzhen Key Laboratory of Synthetic Genomics, Shenzhen Institute of Synthetic Biology, Shenzhen Institutes of Advanced Technology, Chinese Academy of Sciences, Shenzhen, PR China. ✉email: tearroad@163.com; tanghongzhi@sjtu.edu.cn

With the rapid advances in industry and agriculture, serious marine environmental pollution has become an outstanding problem for economic and social development, especially in developing countries[1]. Marine pollution, including heavy metals, petroleum, persistent organic pollutants (POPs), debris, and radionuclides, can directly or indirectly be harmful to living organisms and resources[2]. Plastic pollution has escalated during last 50 years, and estimated contents of plastic in marine are more than 250,000 tons[3]. Removal of microplastics using sorption and filtration has been constructed, such as absorbing the microplastic particles on the surface of marine green algae, filtration of microplastics by membrane technology, and even being combined with membrane bioreactors, with the removal efficiency reaching 97.2%[4]. Two types of *Bacillus* strains isolated from the mangrove sediments were found to degrade different microplastics with the reduction only at 0.0019 mg/day[5].

However, the degradation of microplastics in marine water has few been studied, and only a limited number of bacteria are able to degrade the contaminants under marine conditions with a salinity range between 3.3–3.7%.

Halotolerant bacteria are capable of accumulating high concentrations of various organic osmotic solutes (OOSs), such as compatible solutes (water-soluble sugars or sugar alcohols, other alcohols, amino acids, or their derivatives), ectoine, trehalose, and glycine betaine, etc[6–10]. These OOSs could help maintain an osmotic balance of cytoplasm with the external environment, keeping the normal biological activities for cells. Many marine organisms are slight halophiles (with 3% w/v NaCl in sea water).

*V. natriegens* strain Vmax, isolated from the marine environment, has a generation time of less than 10 min, and cannot grow without NaCl, with an optimal concentration of 2–3% (w/v)[11]. Cassettes containing T7 RNA polymerase gene under the control of either an IPTG- or arabinose-inducible promoter (lacUV5 and araBAD, respectively) were inserted into the large chromosome of *V. natriegens* (ATCC 14048, the original strain of *V. natriegens* Vmax). When expression plasmids containing GFP gene under the control of T7 promoter was introduced, robust GFP expression was detected[12]. However, there have been few reports on using *V. natriegens* as a host for the degradation of environmental pollutants, especially in the marine environment.

In this study, the strain Vmax was used for degrading environmental pollutants under salt stress. First, the salt tolerance mechanisms of strain Vmax were proposed by transcriptomic analysis, and the related salt-induced promoters were identified and characterized. Three models were then constructed to degrade chlorpyrifos (CP), hexabromocyclododecanes (HBCDs), and polyethylene terephthalate (PET), based on the identified promoters. The recycle using for the engineered strains was controlled by binding them to chitin materials, which are specific to *Vibrio* species and abundant in the marine environment.

## Results

**Salt responding mechanism of strain Vmax**. In order to find constitutively expressed elements induced by salt (NaCl or $Na^+$), transcriptomic analysis comparing the treatment group (strain Vmax cultured with 5% (w/v) NaCl) with the control group (strain Vmax cultured with 1% NaCl) was performed. Samples for transcriptomic analysis were taken at the exponential phase, and the gene transcription levels with fold changes in Log2 were clustered in a heat map (Fig. 1a). In total, 1596 genes were identified, of which 728 genes were up-regulated and 868 genes were down-regulated. The up-regulated genes could be divided into 25 categories by GO analysis (Fig. 1b), with the catalytic activity most regulated. The up-regulated genes involved in KEGG metabolic pathway were then divided into six branches, of which the genes related to ABC transporters (ko02010) and two-component systems (ko02020) were the first abundant genes (Fig. 1c).

To identify potential elements associated with salt stress, genes were selected that were up-regulated by more than 4-fold (Supplementary Table 3). Three gene clusters related to salt stress, including the ectoine biosynthesis gene cluster *ectBACD* (up-regulated by 8.96, 9.47, and 8.52-fold), proline/glycine betaine ABC transporters *proWXV* (up-regulated by 9.02, 8.86, and 8.79-fold), and betaine biosynthesis genes *BCCT* (up-regulated by 6.20, 6.09-fold), as well as a cluster regarding flagellin assembly (up-regulated by 7.16, 5.59, 5.43, 4.38, 4.24, and 4.07-fold) were among the 52 up-regulated genes[13]. The $Na^+/H^+$ antiporter typically functions by transporting sodium ions to maintain intracellular osmotic pressure, and its editing gene *NhaC* was up-regulated by 4.49-fold. Based on the transcriptomic analysis results, we propose that the salt tolerance mechanism of strain Vmax involves secreting compatible solutes, such as ectoine or proline/glycine betaine, and enhancing the transport of $Na^+$ via up-regulated transcription levels of genes corresponding to synthesis or channels (Fig. 1d).

**Identification of salt-induced promoters**. Promoters that can be induced by salt stress are important elements for the survival of microorganisms in particular environments. To identify potential promoter-elements induced by salt, the front 400 bp sequence of the most up-regulated genes / gene clusters were selected as candidates. Most of the up-regulated genes were randomly distributed throughout the genome, with the two up-regulated gene clusters *ectBACD* and *proWXV*, in opposite directions. Interestingly, a 717 base pair interval area between the two clusters was identified and was predicted to contain three promoters with two directions (Fig. 2a). Transcription direction of promoter P1 (AAACACT TTATAAAGTCCCTTAACTTCCAGTATGGGGTCCATGTAAT CGT) was matched to gene cluster *proWXV*, and transcription direction of promoters P2 (P2-1: TTCAGAAGCTGTTAATAGCG CGGGGGGATCGTAAATTAGAAAATA ATATAT; P2-2: TAAGG GACTTTATAAAGTGTTTGGTGAGAACCCAGAGCGT GCTT TCTCAC) were the same as for clusters *ectBACD*.

The region containing promoters P1, P2-1, and P2-2 was further characterized by ligating to pS8K-*mRFP*. Highly visible expression of the red fluorescent protein (mRFP) was detected, and the simultaneous two-way activation was measured with an enhanced green fluorescent protein (eGFP) gene (Fig. 2b). Under 3% $Na^+$ stress, the truncated regions, which contain only the promoters P1 (400 bp gene sequence front cluster *proWXV*), P2-1 (reverse 400 bp sequence front gene cluster *ectBACD*), P2-2 (reverse 400 bp sequence front cluster *proWXV*), or the full-length interval area (P12, and its reversed sequence was P21), have transcriptional activity (Fig. 2c). When transcription of *mRFP* gene was initiated with the promoter P2-2, the strongest fluorescence was detected.

**Salt-induced degradation models for CP and HBCDs with promoters P1 and P2-1**. To test and verify the ability of the identified promoters in degradation, three degradation models were constructed and characterized. The main functional genes related to CP degradation were constructed under the control of promoters P1 and P2-1 (Fig. 3a, b), resulting in Vmax-*mpdp12tcpXA*. Here, about 50% of CP with an initial concentration at 100 mg/L could be degraded in 8 h. In addition, the catalysis enzyme system containing a cytochrome monooxygenase (CYP168A1), a 4Fe-4S ferredoxin (Fd), and a NAD(P)H-dependent ferredoxin reductase (FNR) from *P. aeruginosa* HS9 were constructed (Fig. 3c), resulting in strain Vmax-*cyp168A1p12FdFNR*. HBCDs (1 mg/L) could be completely converted to debrominated products,

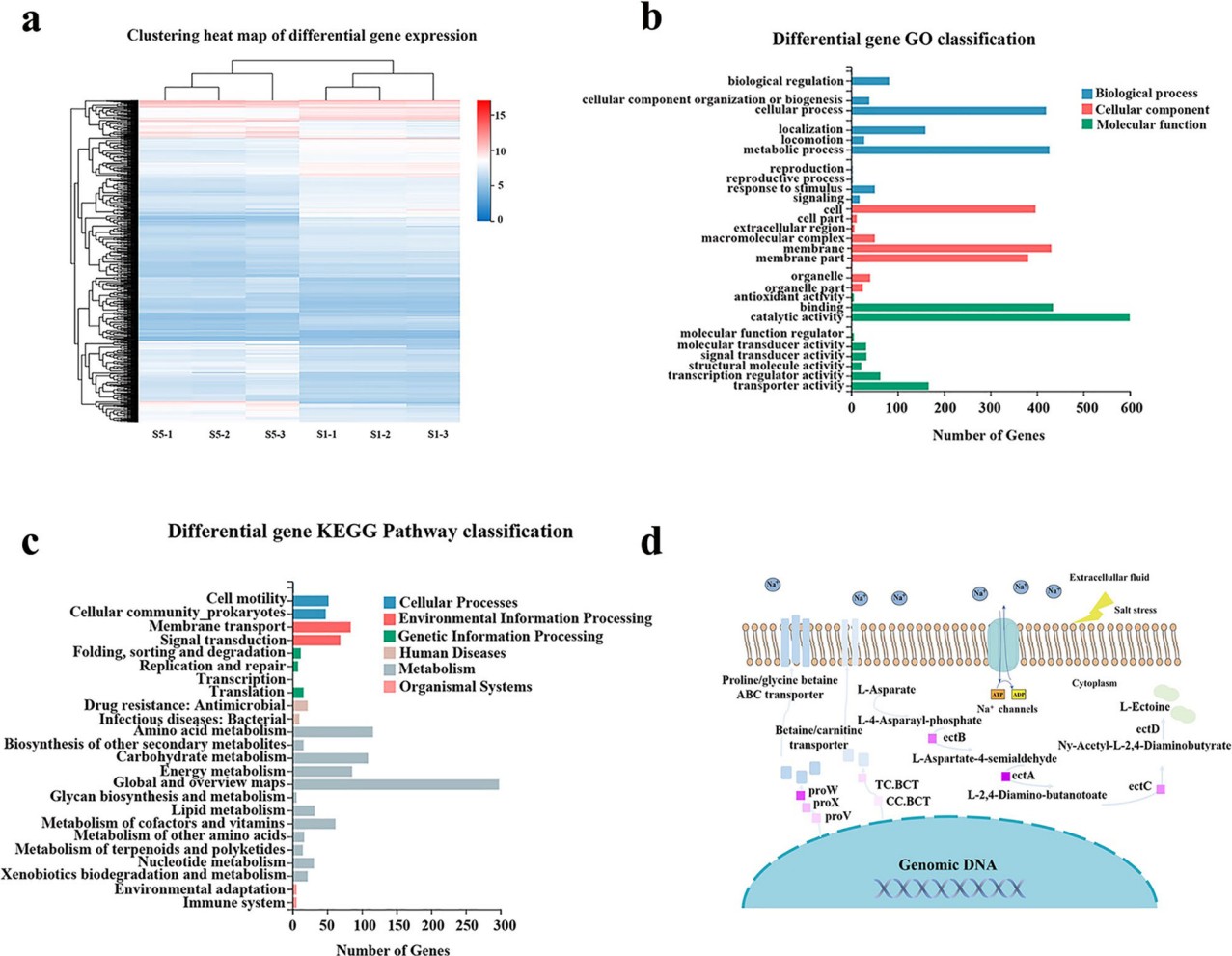

**Fig. 1 Transcriptome data analysis. a** The heat map, based on hierarchical analysis using the log2 (5%/1%) for each gene at two groups (5 and 1%), depicted that 1,596 genes were detected out. The colors were ranging from blue to red, representing the values of log2 (5%/1%). **b** Significantly enriched GO categories for the genes with fold changes in Log2 ≥ 4. **c** Differential gene KEGG pathway classification for the genes with fold changes in Log2 ≥ 4. **d** Proposed halophilic mechanism of strain Vmax based on the transcriptome analysis.

including pentabromocyclododecanols (PBCDOHs) and tetra-bromocyclododecadiols (TBCDDOHs) within 8 h (Fig. 3d).

**Salt-induced degradation models for PET with promoters P1 and P2-2.** To enhance the efficiency of transcription, a third model was constructed with the promoter area shortened from P1 to P2-2. As PETase or LCC could catalyze PET to mono-(2-hydroxyethyl) terephthalic acid (MHET) and bis-(2-hydroxyethyl) terephthalic acid (BHET) as main products, then the MHETase or Tfca can catalyze BHET and MHET further to ethylene glycol (EG), terephthalic acid (TPA), and other components under mild conditions (Fig. 4a). Here, four engineered PET hydrolases with the highest catalytic activity were chosen to construct salt-induced PET degrading strains. The degradation of PET was confirmed via generation of the hydroxylation products BHET and MHET. Both whole cells and crude cells could degrade PET, and the degradation efficiencies differed (Fig. 4b, c). For whole cells, the degradation rates of the four engineered constructs ranked as Vmax-*MHETaseP12₂PETase* (MPP) > Vmax-*PETaseP12₂MHETase* (PPM) > Vmax-*TfcaP12₂LCC* (TPL) > Vmax-*LCCP122Tfca* (LPT). The products BHET and MHET were detected with a maximum accumulation at 10 mg/L and 11 mg/L after 8 d in the MPP constructs (Fig. 4b). For the crude enzymes of the PPM constructs, BHET and MHET accumulated to 5.0 and 40 mg/L in the first 24 h,

and decreased to 0 at 48 h, respectively (Fig. 4c). For the crude enzymes of the MPP constructs, BHET and MHET accumulated to 127 and 14 mg/L in the first 24 h and 60 h, respectively. For the crude enzymes of the LPT constructs, BHET and MHET accumulated to 105 mg/L and 48 mg/L in the first 24 h. For the crude enzymes of the TPL constructs, BHET accumulated to 128 mg/L in 120 h, while MHET accumulated to 76.5 mg/L by 24 h, and then decreased to 11 mg/L at 120 h. The activities for crude enzymes were stronger than activities of the whole cells, owing to the suitable temperature for enzymes. Changes in the surface morphology of PET membrane samples, treated by strains PPM, MPP, LPT, and TPL, were shown in Fig. 4d. Compared to untreated PET, varying degrees of fragmentation were detected for the treated samples. For the PET membrane degradation, the most obvious change was observed in the MPP samples, which matched to the accumulation of degrading products.

**Environmental and ecological safety of engineered bacteria.** To achieve better application of engineered strains in environmental bioremediation, prevention of engineered strain leakage is essential for environmental and ecological safety. The chitin-binding protein GbpA of *Vibrio cholerae* was added to the engineered Vmax strains (Vmax- *cyp168A1p12FdFNR*) in the pMD18T vector, resulting in Vmax-*GbpA-cyp168A1p12FdFNR* (*VgHBCD*). In this modified

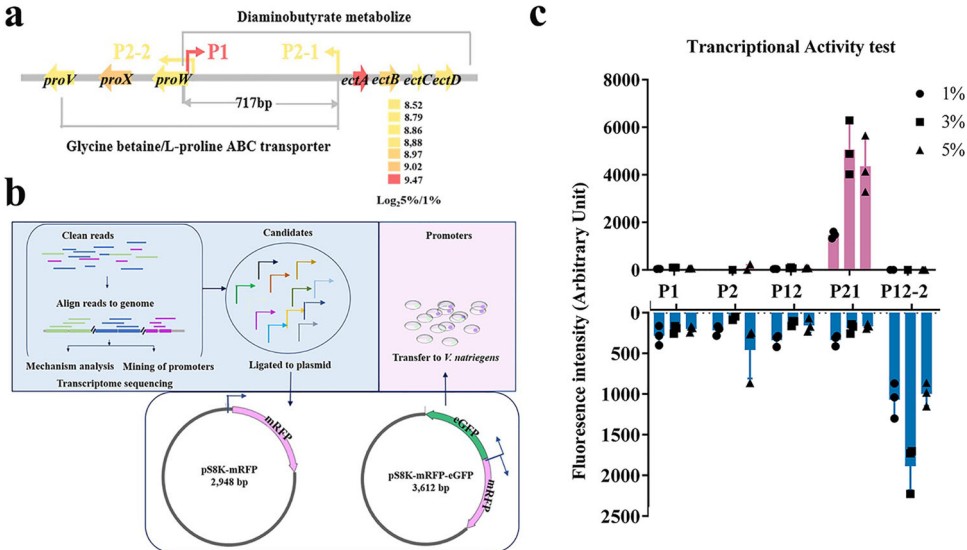

**Fig. 2 Identification of salt-induced promoters. a** Schematic of two up-regulated gene clusters, with three proposed promoters. (Color represents the up-regulation of the response genes; the sequences denote the proposed area for promoters, and the magnified letters represent the start site for transcription). **b** Flow chart of the identification and characterization of the proposed promoters. **c** Detection of the fluorescence intensity for mRFP/eGFP in the unit cell. P1 (400 bp gene sequence front cluster *proWXV*), P2-1 (reverse 400 bp sequence front gene cluster *ectBACD*), or P2-2 (reverse 400 bp sequence front cluster *proWXV*), and the full-length interval area (P12, and its reversed sequence was P21). Error bars are standard error of mean.

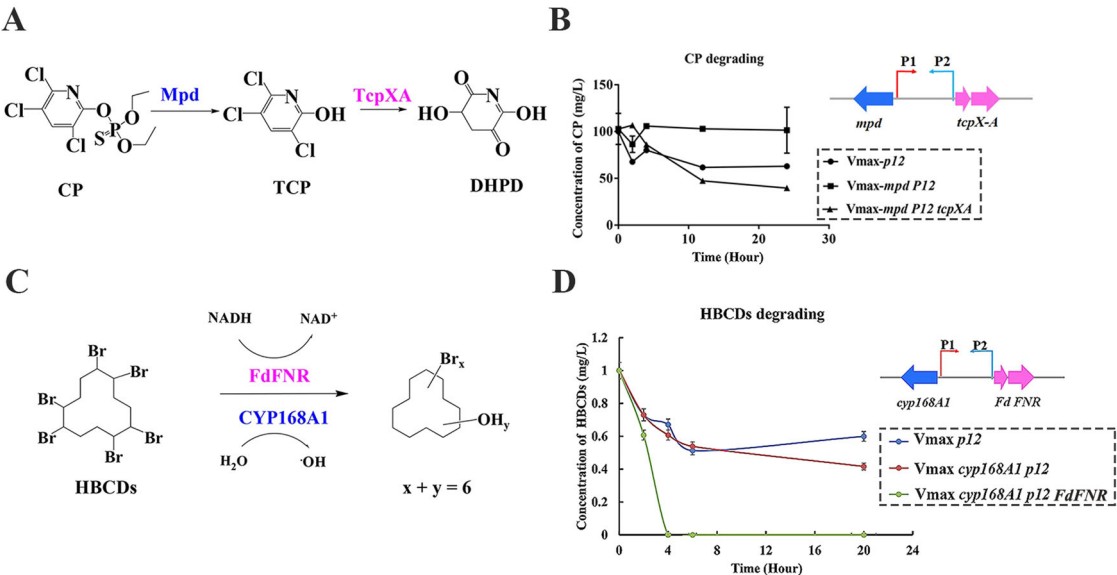

**Fig. 3 Construction and characterization of the salt-induced model with promoters P1 and P2-1. a** Research basis of chlorpyrifos (CP) degradation and the plasmid diagram for salt-induced CP degradation. **b** Verification of the metabolic efficiency of the pesticide CP. **c** Research basis of hexabromocyclododecane (HBCD) degradation and the plasmid diagram for salt-induced HBCD degradation. **d** Verification of the metabolic efficiency of the HBCDs.

strain, a 53 kDa protein was expressed and detected by SDS-PAGE, which matched to the molecular weight of protein GbpA, and the adhesion of *VgHBCD* to chitin was enhanced about 2-fold (Fig. 5a, b). Biofilm could be observed under scanning electron microscope (SEM) on the surface of chitin (Fig. 5c). When the *VgHBCD*-adhered chitin material was washed with NSS buffer to test the recovery potential, about 80% of the bacterial biofilm was retained on the surface of chitin. In addition, the HBCDs degradation ability of *VgHBCD* when bound to chitin was detected after each wash.

The HBCDs degradation rates were 50% at first turn and 100% at the second and third turns (Fig. 5d). The recombinant strain *VgCP* binding to chitin could completely degrade 100 mg/L chlorpyrifos all three times recycle (Fig. 5e). Strains *VgPPM*,

*VgMPP*, *VgLPT*, and *VgTPL* binding to chitin could degrade mPET, and the accumulated products BHET and MHET were detected by HPLC. In the first turn, the products BHET and MHET accumulated substantially, and more faintly in the second and third turns (Fig. 5f).

## Discussion
Due to industrial activities, organic-contaminated environments are frequently experience saline and hypersaline stress. High salt concentrations (>1%, w/v) may cause a loss of microbial activity, limiting enzymatic activity. Strategies to overcome these problems include adapting bacteria to high salinity and using

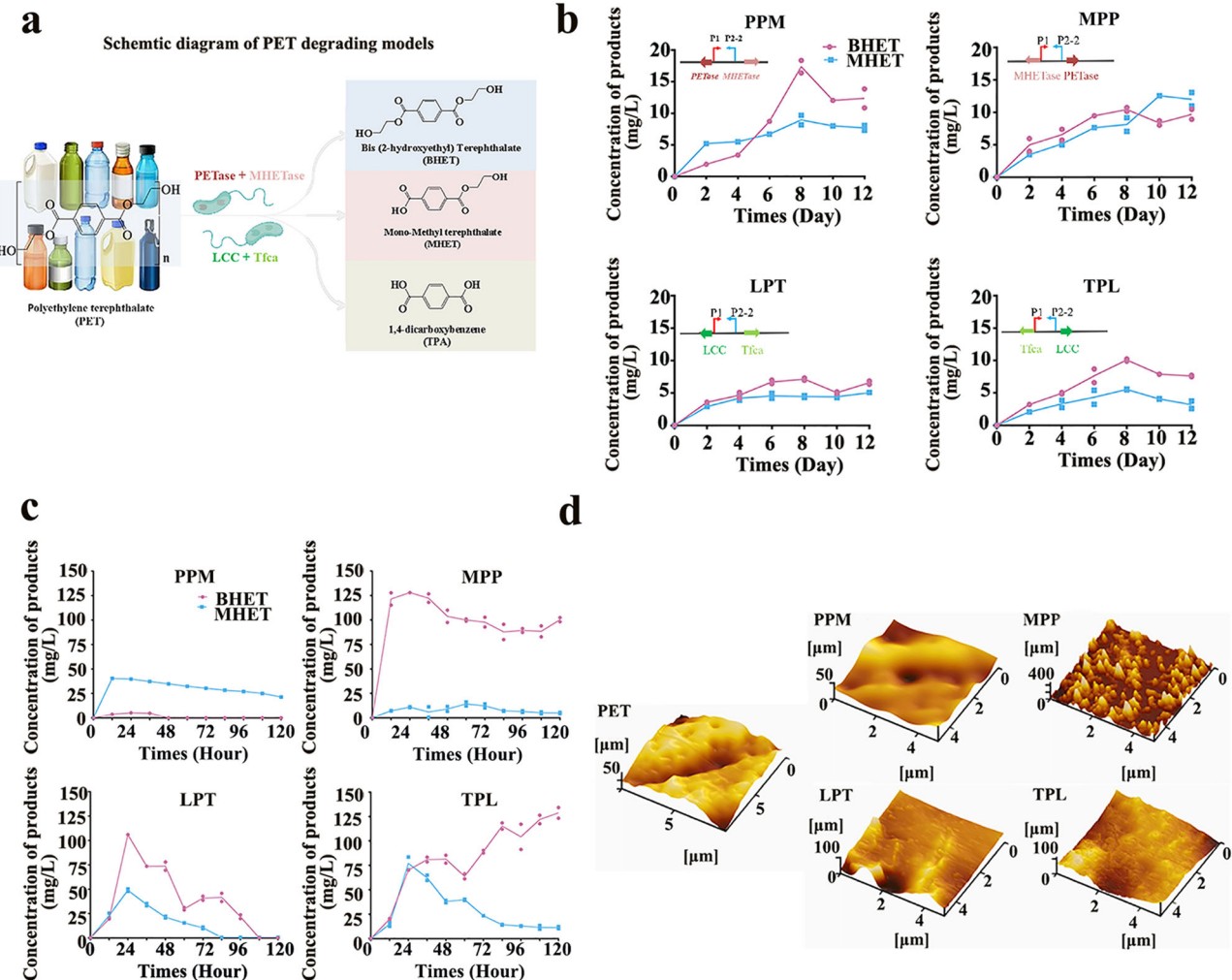

**Fig. 4 Construction and characterization of the salt-induced model with promoters P1 and P2-2. a** Combination of PET hydrolases for polyethylene terephthalate (PET) degradation in this work: PETase + MHETase or LCC + Tfca. **b** Accumulation of products mono-(2-hydroxyethyl) terephthalic acid (MHET) and bis-(2-hydroxyethyl) terephthalic acid (BHET) in the four engineered PET strains' whole cell degrading systems. Maps of the constructed plasmid are drawn in gray. **c** Accumulation of products MHET and BHET in the four engineered PET degradation strains' crude enzyme systems. **d** Changes of the surface morphology of PET membrane samples verified by an atomic force microscope (AFM) (NANOCUTE II, Seiko Instruments Inc.).

microorganisms with high salt tolerance. Limited information about the salt responding mechanism of *V. natriegens* has been reported, including its utility as a platform to degrade organic pollutants.

The salt responding mechanisms of various halophilic and halotolerant microorganisms have been extensively researched, and include selectively transporting the inorganic ions $Na^+/K^+$ in the cytoplasm to the accumulation of specific organic substances of low-molecular weight, such as ectoine, proline, betaine, trehalose, choline-O-sulfuric acid, and carnitine. Information regarding the transcriptional regulation of biosynthesis is overwhelmingly about the biosynthesis of ectoine[14]. Most of the reported salt tolerant bacteria could secrete compatible small molecules to adapt to the salt stress, such as *Methylotuvimicrobium alcaliphilum* 20Z (NC_016112.1)[15], *Bacillus halodurans* C-125 (BA000004.3)[16], *Streptomyces coelicolor* A3 (AL645882.2)[17], *Halomonas elongata* HEK1 (FN869568.2)[18], and *Chromohalobacterium salexigens* DSM 3043 (CP000285.1)[19]. Ectoine biosynthesis genes (*ectABC* cluster), located in the genome of the above bacteria, are transcribed under the control of the salt-induced promoters adjacent to the "ATG" codon of *ectA*, such as *ectAp1*, *PectA*, *PectA*, *PectA1-4* and *PectB*, and *PectA1-4* and *PectB*. In *V. natriegens*, the *ectABC* gene cluster

was adjacent to the glycine betaine/proline transport system genes, with the opposite direction, under the control of three promoters (Supplementary Fig. 1). The location of genes related to salt stress in strain Vmax presents an obviously location-specific advantage, compared to the other reported halophilic or halotolerant microorganisms. The genes related to $Na^+/K^+$ transcription and ectoine, proline, and betaine biosynthesis could be induced to express nearly at the same time. As a result, that the reduced distance between the two gene clusters for ectoine synthesis and betaine/proline transportation in *V. natriegens* compared to other halophilic and halotolerant bacteria, as well as the sensitive promoters, improved its ability to maintain osmotic balance.

Considering the necessity of bioremediation in salt-affected environments, enhancement of the growth and degradation ability of specialized bacteria under salt stress is meaningful. Thus, it is of utmost importance to evaluate the potential of the identified promoters for the practical application of bioremediation in organic-contaminated environments with salt stress. Here, three degradation models were constructed. Chlorpyrifos (CP) is one of the most widely used organophosphorus (OP) pesticides, causing neurobehavioral damage. The degradation rate of Vmax-*mpdp12tcpXA* (50 mg/L in 24 h) was much lower than that of the

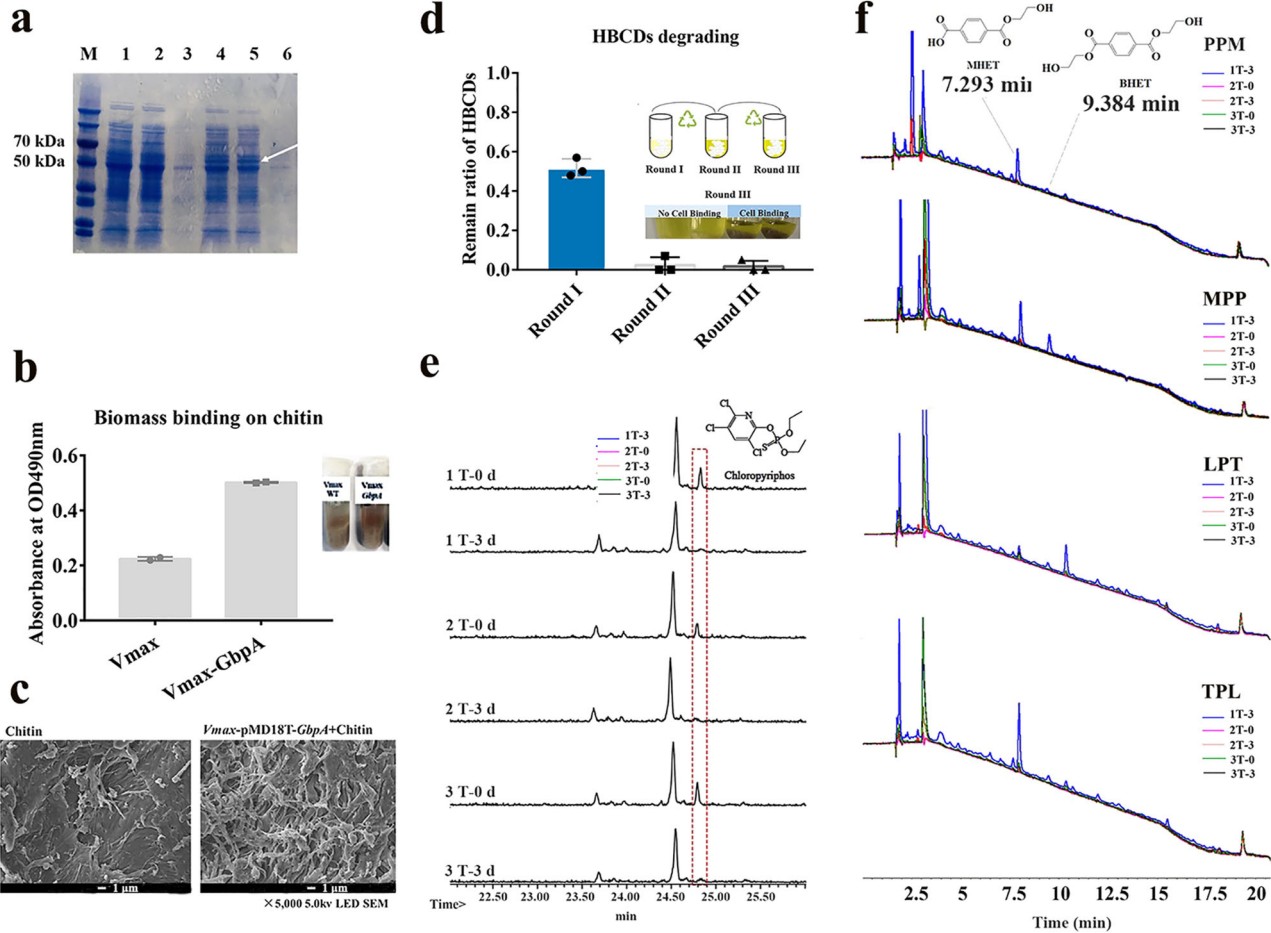

**Fig. 5 Environmental and ecological safety. a** Analysis of the expression of the chitin-binding protein GbpA by SDS-PAGE. M, protein marker; lane 1–3, crude enzyme, supernatant, precipitate of strain Vmax; lane 4–6, crude enzyme, supernatant, precipitate of strain Vmax-GbpA. **b** Measurement of biomass binding on chitin by a 3-(4,5)-dimethylthiahiazo (-z-y1)-3,5-di- phenytetrazoliumromide (MTT) assay kit. Error bars are standard error of mean. **c** Morphology of the bio-anchored chitin under scanning electron microscope (SEM) S3400II (Hitachi, Japan) with an accelerating voltage of 20 kV. **d–f** Verification of the recovery rate and degradation effect of the chitin-fixed strains for HBCDs, CP, and mPET, respectively.

natural degrading bacterium *Stenotrophomonas* sp. YC-1, which can degrade 100 mg/L CP in 18 h. It is likely that the expression of genes under the control of P1 were not sufficient to metabolize the first product, 3, 5, 6-trichloro-2-pyridinol (TCP)[20,21]. In the second model, hexabromocyclododecanes (HBCDs), which are the second most commonly used brominated flame retardants (BFRs) added in building materials, electronics, textiles, and plastics, were selected as the target substrate. In the degradation process of halogenated organic compounds, the initial dehalogenation is the most crucial step. Here, the degradation rate of HBCDs was enhanced about 10-fold compared to strain HS9, with a degradation rate of 0.64 mg/L over 14 days. The HBCD degradation rate of Vmax-*cyp168A1p12FdFNR* was much faster than the degrading rate of HS9[22,23]. Owing to the efficiency of the electron supply in the salt-induced HBCDs degrading model, containing the catalysis enzyme CYP168A1 and the electron suppling enzymes ferredoxin (Fd) and ferredoxin reductase (FNR), the degrading rate of HBCDs was robustly enhanced. In the current study, the marine microorganism *V. natriegens* was used as a host to express catalysis enzymes dependent on an environmental factor for growth (Na$^+$). The usage of gene-edited bacteria in the environment may suffer some biosecurity risks. They may refer to unauthorized access, loss, theft, misuse, diversion or intentional release. If a biosecurity accident happens, it would pose a huge threat to humans and nature[24]. Our studies

help to address the challenges (insufficient robustness of laboratory strains, expression dependent on chemical inducers) in constructing genetically engineered bacteria intended for environmental release.

The problem of marine microplastics pollution has attracted widespread attention, ranking as one of the top ten emerging environmental problems globally. Owing to their small size (diameter < 5 mm), wide range of sources, large quantities, and release of additives, such as metals and toxic organic pollutants, these microplastics result in significant biological toxicity. Plastic waste is widespread in offshore, marine waters, and sediments, and continuously accumulates[25]. Although many kinds of PET hydrolases have been investigated in various microorganisms, including cutinase, lipase, and PETase. The applications of those enzymes to engineering bacteria degrading PET in the marine environment are limit. However, the immobilized cells may exhibit improved catalytic activities, shorten bioremediation time, lower production cost with lower biosecurity risks[26]. In our study, in model three, strains PPM, MPP, LPT, and TPL could degrade PET in 0.5 × NSS, indicating that these recombinant *V. natriegens* strains could be used in marine environments to bioremediate PET pollution.

This study helps address many concerns about the low growth and degradation rates of natural microorganisms used for bioremediation under high salt stress in marine environment. Marine

*V. natriegens* should have good potential for bioremediation of the polluted marine environment.

## Materials and methods

**Materials**. Chlorpyrifos (CP, ≥99% analytical grade), 3,5,6-trichloro-2-pyridinol (TCP, ≥99% analytical grade), polyethylene terephthalate (PET), bis-(2-hydroxyethyl) terephthalic acid (BHET), and mono-(2-hydroxyethyl) terephthalic acid (MHET) were purchased from Meryer (Shanghai, China). 1, 2, 5, 6, 9, 10-Hexabromocyclododecanes (HBCDs, ≥95% analytical grade) were purchased from Anpel Laboratory Technologies (Shanghai, China). Ethyl acetate, methanol, and all other regents and solvents used in this study were of analytical grade.

**Strains and culture media**. *V. natriegens* strain Vmax was purchased from Synthetic Genomic Company (Calipatria, CA). *Pseudomonas aeruginosa* HS9 was obtained from the lab store[22]. The minimal salts medium (MSM) used in this work contained (per liter) 13.3 g Na$_2$HPO$_4$·12H$_2$O, 4 g KH$_2$PO$_4$, 0.2 g MgSO$_4$, 2.0 g NH$_4$Cl, 9.5 g NaCl, and 0.5 mL of trace elements store solution. Trace elements store solution consisted of (per liter) 0.05 g CaCl$_2$·2H$_2$O, 0.05 g CuCl$_2$·2H$_2$O, 0.008 g MnSO$_4$·H$_2$O, 0.004 g FeS·7H$_2$O, 0.1 g ZnSO$_4$, 0.1 g Na$_2$MuO$_4$·2H$_2$O, and 0.05 g K$_2$WuO$_4$·2H$_2$O, and pH 7.0. The nigh salts solution medium (NSS) contained (per liter) 8.8 g NaCl, 0.735 g Na$_2$SO$_4$, 0.125 g KCl, 0.02 g KBr, 0.935 g MgCl$_2$·6H$_2$O, 0.205 g CaCl$_2$·2H$_2$O, 0.004 g SrCl$_2$·6H$_2$O, and 0.004 g H$_3$BO$_3$[27].

**Transcriptomic analysis**. Transcriptomic analysis was conducted by comparing the transcription profile of the cells incubated in media with 1 and 5% (w/v) final concentration of NaCl. Strain Vmax was cultured in 2 L flasks containing 1 L of 5% NaCl Luria-Bertani broth (LB5). For the control group, strain Vmax was grown in lysogeny broth (LB). All the samples were cultured in 30 °C thermostat shakers at 200 rpm. Cells were harvested when OD600 reached to 0.6~0.8 for transcriptome sequencing, frozen by liquid nitrogen and stored at −80 °C before sequencing. The extracted total RNA was detected by the DNBSEQ platform (BIG, Shenzhen, China). Genes with high transcription levels under a high salt concentration (5% NaCl) (up-regulation fold change ≥4) were summarized as the research candidates. Clean-reads were matched to the genome sequence by the Hierarchical Indexing for Spliced Alignment of Transcripts (HISAT) program[28,29].

**Activity determination of promoters**. To determine the activity of the proposed promoters (front 400 bp sequence), the interval area was ligated to clone the vector pSK8k-*mRFP* between the *PstI* and *EcoRI* sites. The front sequences were amplified from Vmax genomic DNA, using the primers described in Supplementary Table 1. The purified fragments were ligated to linear pSK8k-*mRFP* by using a 2× CloneExpress MultiS One Step Cloning Kit (Vazyme, Nanjing, China). The regions containing promoters P1 and P2 were further verified by ligating to pS8K-*mRFP*. The resulting vectors were transferred to strain Vmax by electro-transformation and the competent cells were prepared as following, strain Vmax was grown in 3% LB, cells were cultured in 30 °C thermostat shakers at 200 rpm, and harvested when OD600 reached to 0.6–0.8, the cells were washed with the wash buffer (7 mM K$_2$HPO$_3$ and 680 mM sucrose) for twice, and resuspended with wash buffer[8]. The fluorescence intensity of GFP and RFP was determined using a multimode microplate reader 20 M (Tecan & Spark, Switzerland) with excitation at 510 nm and reading the emission at 485 nm for GFP and with excitation at 607 nm and reading the emission at 584 nm for RFP[30,31]. The cell growth was measured at 600 nm. The unit activity of the promoters was calculated by:

$$\text{Activity} = [\text{Fluorescence intensity(sample)} - \text{Fluorescence intensity(control)}]/\text{OD600}.$$

**Construction of degradation models**. The codon optimized genes involved in the chlorpyrifos (CP) metabolization were synthesized by GENEWIZ Company (Suzhou, China). The accession numbers of the CP degrading genes *mpd*, *tcpX*, *tcpA*, *dhpI*, and *dhpJ* were ABD92793.1, AGC65457.1, AGC65458.1, KC294623.1 (2341–3056), and KC294623.1 (3081–3959), respectively[20,21]. In the HBCDs degradation model, gene *cyp168A1*, a 4Fe-4S ferredoxin (Fd), and a NAD(P)H-dependent ferredoxin reductase (FNR) were amplified from *P. aeruginosa* HS9. The PET hydrolases PETase, MHETase, LCC, and Tfca were synthesized with the sequences from references[32–35] as templates by GENEWIZ Company. All the primers used are listed in Supplementary Table 2. The CP/HBCDs degradation pathways were positioned in the clone vector pAMmcs between *EcoRI* and *XhoI* sites. The reverse gene *mpd/ cyp168A1* was under the control of promoter P2-1, while the other genes (*tcpXA-dhpIJ/FdFNR*) were controlled by promoter P1, resulting in *pAM-mpdp12tcpXA* and *pAM-cyp168A1p12FdFNR*, respectively. The PET hydrolases, PETase[32], MHETase[33], LCC[34] and Tfca[35], were constructed in the same vector pSK8k as the promoter activity identification, under the control of promoters P1 and P2-2.

After electro-transformation to strain Vmax, single clones were verified by PCR, obtaining Vmax-*mpdp12tcpXA*, Vmax-*cyp168A1p12FdFNR*, Vmax-*PETaseP12MHETase* (PPM), Vmax-*MHETaseP12PETase* (MPP), Vmax-*LCCP12Tfca* (LPT), and Vmax-*TfcaP12LCC* (TPL) constructs. To determine their ability to degrade PET, CP, or HBCDs, the engineered strains grown to the

logarithmic phase in 5% NaCl lysogeny broth (LB5) were collected and washed with the 0.5× NSS medium three times, and resuspended with 0.5× NSS. The final optical density was adjusted to an OD600 of 1.0. One-gram PET (containing 0.5 g micro-PET, and 0.5 g PET membrane), 100 mg/L CP, or 1 mg/L HBCDs were added to 20 mL of the cell lysate as the substrate, separately, all the reactions were carried out at 30 °C. One mL reaction mixture was extracted from the reaction system every 2 days. Hydrochloric acid (1%) was added to samples to terminate the reaction. All samples were stored at −80 °C until use, and three biologically independent samples were detected for each single point[36,37]. For the crude enzyme activity tests, the resuspended cell suspension was broken by high pressure homogenizer (APV-2000, Germany) at 4 °C, and then the cell debris was removed by centrifugation at 10,000 rpm for 40 min. One-gram PET was added to 20 mL of the supernatant as the substrate, and the reaction with PPM, MPP, LPT, and TPL constructs were carried out with the thermostatic shakers at 44 °C and 55 °C, respectively. Also, 1 mL reaction mixture was extracted from the reaction system every 2 h, the samples were prepared and detected with the same methods as the whole cell degrading samples.

**Immobilization and continuous recycling**. Immobilization of the engineered Vmax strains on chitin was conducted to avoid biological leakage and enable continual recycling of the degrading micro-resource[38,39]. The gene (Accession No. CP047296.1, 755825–757282) encoding chitin binding protein GbpA was amplified from *Vibrio cholerae*[40,41], with primer GbpAF: 5'-ATGAAAAAACAACCT-3', GbpAR: 5'-TTAACGTTTATCCCACG-3'. The amplified product was ligated into pMD18T-Blunt Vector (TransGen Biotech, China) and then transformed into *E. coli* Top10[42]. The transformants were sequenced using the universal primers M13F-GbpAF or M13R-GbpAR after plasmid extraction. The resulting vector, pMD18T-*GbpA*, was transferred to strain Vmax-*mpdp12tcpXA*, Vmax-*cyp168A1p12FdFNR*, Vmax-*PETaseP12MHETase* (PPM), Vmax-*MHETaseP12PETase* (MPP), Vmax-*LCCP12Tfca* (LPT), and Vmax-*TfcaP12LCC* (TPL), respectively, forming Vmax-*GbpA-mpdp12tcpXA* (*VgCP*), Vmax-*GbpA-cyp168A1p12FdFNR* (*VgHBCD*), Vmax-*GbpA-PETaseP12MHETase* (*VgPPM*), Vmax-*GbpA-MHETaseP12PETase* (*VgMPP*), Vmax-*GbpA-LCCP12Tfca* (*VgLPT*), and Vmax-*GbpA-TfcaP12LCC* (*VgTPL*) constructs.

Engineered strains *Vg* were grown to the logarithmic phase in LB5, collected, and washed with the NSS medium three times. Then, 2 mL of cell pellet was resuspended with 350 mL NSS, supplemented with 100 mg/L CP, 10 mg/L HBCDs, and 0.1 g micro-PET (mPET). The resulting suspension was added to 0.3 g of sterilized chitin (CAS: 1398-61-4) (Sigma-Aldrich, America) in a 5 mL glass bottle, with 1 mL 0.5 × NSS buffer was added to the reaction system. The binding reaction was stationary incubated at 30 °C for 3 d. To test the efficiency of recycled bacteria, the engendered HBCDs degrading strain was taken as module to calculate the HBCDs remaining rates for three times, by detecting the remaining HBCDs concentrations. The solid-chitin-binding-bacteria was washed three times with the NSS medium for recycle using (centrifugation at 3000 rpm for 2 min). Due to the function of chitin binding protein GbpA, the efficiency of engineered strains *Vg* binding to chitin was enhanced. The biomass binding on chitin was measured via a (3-(4,5)-dimethylthiazol-2-yl)-2,5-diphenyltetrazoliumbromide (MTT) assay kit. The strain was incubated and added with 100 µL MTT (5 g/L) and kept in the dark for 4~5 h, following that 100 µL DMSO was added and read at 570 nm. The mortality percentage was calculated[43]. Concentrations of the added substrates were detected with same method as in Section 2.5.

**Metabolite identification and analytical methods**. Samples for CP and HBCDs degradation were prepared by first adding NaCl to the stored samples in excess. Ethyl acetate was then added at a volume ratio of 1:1 to extract the substrate. Then, the mixture was vortexed for 30 s with a vortex oscillator (YETO, vortex-2, China) followed by centrifugation at 12,000 rpm for 5 min. The organic phase was used for detection. Samples for PET degradation were prepared by adding 20% dimethyl sulfoxide (DMSO), followed by the extraction as used for the CP and HBCD samples. The concentration and intermediate metabolites of CP were detected by gas chromatography-mass spectrometry (GC-MS) (Agilent & GC-7890B; MS-5977B) detection. The organic phase was incubated with an equal volume of BSTFA at 70 °C for 30 min before injection[44].

HBCDs were quantified by ultra-performance liquid chromatography-quadrupole time-of-flight mass spectrometry (UPLC-TOF/MS) equipped with an Eclipse XDB C18 analytical column (5 µm, 4.6 × 150 µm, Keystone Scientific, Agilent). A mobile phase of water and methanol at a flow rate of 0.25 mL/min was applied for the target compounds. The proportional gradient of the mobile phase was started at 95% methanol, and increased linearly to 100% over 25 min, then decreased immediately to 95% and held for 10 min. For mass spectrometry analysis, the ionization source was run in negative mode, and MS detection was set from *m/z* 0 to 1,700. All target compounds were extracted based on their hydrogen adduct ions [M + H]$^-$ at *m/z* and characterization of the bromine isotope.

Bis-(2-hydroxyethyl) terephthalic acid (BHET) and mono-(2-hydroxyethyl) terephthalic acid (MHET) were quantified by ultra-performance liquid chromatography (HPLC) equipped with an Eclipse XDB C18 analytical column (5 µm, 4.6 × 150 µm, Keystone Scientific, Agilent). A mobile phase of water with 0.1% formic acid and methanol at a flow rate of 0.8 mL/min was applied for the target compounds. The

proportional gradient of the mobile phase was started at 5% methanol, and increased linearly to 44% over 12 min, then increased linearly to 70% over 3 min and held for an additional 3 min before returning to 5% methanol immediately.

**Analysis of surface morphology of the materials**. The morphology of the bio-anchored chitin was observed and analyzed by scanning electron microscope (SEM) S3400II (Hitachi, Japan) with an accelerating voltage of 20 kV. After being coated with gold, all samples were pasted on the SEM sample plate and observed separately. Changes in the appearance of PET membrane surfaces were measured using an atomic force microscope (AFM) (NANOCUTE II, Seiko Instruments Inc.), with average size of 10 nm. The PET samples were washed with distilled water three times, and then washed twice with ethanol.

**Statistics and reproducibility**. All assays were performed in duplicates and repeated in least three independent experiments. Concentration-degradation-curves were generated with GraphPad Prism 8.0 software.

**Reporting summary**. Further information about research design is available in the Nature Research Reporting Summary linked to this article.

## Data availability

All data supporting the findings of this study are available within the article and its Supplementary Information file (Supplementary Fig. 1, and Supplementary Tables 1–3). Additional information, relevant data and unique biological materials will be available from the corresponding author upon reasonable request.

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

## Acknowledgements

This work was supported by a grant (2021YFA0909500) from National Key R&D Program of China, by the grant (32030004) from National Natural Science Foundation of China, by the grant (20XD1421900) of Shanghai Excellent Academic Leaders Program, and by the grant (17SG09) of Shuguang Program from Shanghai Education Development Foundation and the Shanghai Municipal Education Commission.

## Author contributions

L.H. and H.T. conceived and designed experiments. L.H. performed experiments. H.T. and P.X. received projects and contributed reagents and materials. L.H. and H.T. wrote the paper. L.H., J.N., H.T., C.Z., J.D., and P.X. discussed and revised the manuscript. All authors commented on the manuscript before submission. All authors read and approved the final manuscript.

## Competing interests

The authors declare no competing interests.
