## [Peer Review File · Communications Biology]

Reviewers' comments:

Reviewer #1 (Remarks to the Author):

In this study, authors tried to find functional promoters in high salt stress and used in marine pollutant degradation. This is a meaningful basic research to solve the marine pollution including plastics, organophosphorus pesticides and halogenated organic compounds. However, some flaws in the manuscript are present and need to be fixed.

LINE 257-258: If the control group and treatment group reached exponential phase at the same time? Has the growth curve made for this strain when cultivated with different concentration of NaCl? How long have you cultivated these groups and when did you determine their transcription?

LINE 258: The de color bar should be marked clearly some detailed information, such as 'relative expression', 'log2 fold' or 'xxx (brief information of treatment group) vs xxx (brief information of control group)'..

LINE 261: 'catalytic activity' in Figure 1B should be 'catalytic activity'?

LINE 262: Which seven branches? Do you mean cellular process, environmental information processing, genetic information processing, human diseases, metabolism and organismal systems? If yes, it should be six.

LINE 263: Do authors mean that genes from 'global an overview maps' occupied the largest proportion? The global maps (map numbers 01100s) and the overview maps (map numbers 01200s) are a special class of metabolic pathway maps in the KEGG PATHWAY database, presenting global and overall pictures of metabolism. They contain a large number of genes and enzymes from diverse pathways instead of a specific metabolism. So, what is the significance of this conclusion?

LINE 265: third?

LINE 267: why did the authors use 4-fold as the cut-off?

LINE 267-273: Any references provided evidence to prove these 3 gene clusters were related to salt stress?

LINE 273-275: References are needed.

LINE 276: Vmax should be 'V. natriegens Vmax' or 'strain Vmax', similar mistakes in the manuscript should be corrected.

LINE 266-279: References are needed.

LINE 286: Did the authors mean that the direction of these two genes was opposite? Up-regulated opposite gene clusters?

LINE 289: Names of genes or gene clusters should be marked clearly in Figure 2A.

LINE 301: The unit of fluorescence intensity (y-axis) was missed.

LINE 302: Which P2 promoter? P2-2 or P2-1? In Figure 2C, which one was the promoter P2-1 or P2-2 used? And the legend of figure 2 should be more detailed. What did the P1 P2 P21 P12 mean in Figure 2C?

LINE 309: why not use the P2-2? Any result showed that P2-1 was better than P2-2?

LINE 313: What did the 'x+y=6' mean in Figure 3C?

Figure 3D: 1.0

LINE 320-322: The 'PET hydrolase' mentioned here is PETase? These products might be generated by PETase and MHETase. PETase cleaves BHET to MHET and EG, and the soluble MHET product is further hydrolyzed by MHETase to produce TPA and EG. And references are needed here.

LINE 329: What are the differences between MPP and PPM? Exchanged the location of promoters? But the gene cluster of PPM and MPP shown in Figure 4B seems same.

Figure 5B: missing the unit of the Y-axis. It seems that the error bars were missed. Any repeats about this test?

LINE 566: 'Environ Sci Technol' should be italic as above.

LINE 568-569: The format of this reference should be unified.

Figure 5A: The information of the sample in each lane should be detailedly described in the legend.

Figure 5C: The scale was missed.

Reviewer #2 (Remarks to the Author):

Bioremediation of marine pollutants like microplastics, pesticides, or halogenated additives like commonly used brominated flame retardants under high salt conditions is an exciting, relevant subject which would be of high interest to the readership of Communications Biology and comply entirely with the journal profile. Even though many articles have been published concerning the topics, especially compiling aspects of new discovered high salt-induced functional promoters, the rapidly growing marine bacterium *Vibrio natriegens* strain Vmax with high potential as a biotechnological production platform, highly relevant pollutant degradation, and the sustainable strategy of biomass recycling of the biocatalyst could make a significant contribution. The manuscript is written in an engaging and lively style. Still, it would benefit from some persuasive proofreading, recent comprehensive references and detailed data analysis and interpretation, and more explicit, compelling arguments in the results and discussion parts as parts of the manuscript made it challenging to follow.

My recommendation to the editor is to reject this paper. In my opinion, the essential revisions are too fundamental for the submission to continue being considered in its current form. As a research article, the manuscript should provide an overview of the published literature, significant advances in the genetics of halophilic microbes, biodegradation of microplastics, other xenobiotics (e.g. pesticides, flame retardants), their combined application and presentation in technically sound data. The latter is missing in many aspects of the current paper, but especially in the last two chapters of the Results section: Salt-induced degradation models for PET and the Environmental and ecological safety of engineered bacteria. Many aspects of the results are poorly discussed or completely ignored, like the biomass immobilization on chitin, the biocatalyst recycling and the title announced feature of controllable and environmental safe application of GMOs is missing.

Additionally, as a reader and reviewer, the current paper lacks originality (PET-biodegradation in marine environments has been a focus of many recently published papers; same for HBCD and chlorpyrifos), the presentation needs significant improvements, and the potential impact is low. I would strongly advise the authors to re-write their introduction, discussion and conclusions. The chapters concerning halophilic responses in *V. natriegens*, salt-induced promoters, and biodegradation are inspiring and should be enhanced by providing more details. The data regarding mPET-degradation are attractive as well. Still, the lack of consistency, especially the density of data points in the first period of 0h – 72h, is unfortunate. These experiments should be reiterated for the best of this manuscript.

Please see the attachment for Reviewer #2's line by line comments.

Response to Reviewer 1

In this study, authors tried to find functional promoters in high salt stress and used in marine pollutant degradation. This is a meaningful basic research to solve the marine pollution including plastics, organophosphorus pesticides and halogenated organic compounds. However, some flaws in the manuscript are present and need to be fixed.

Response: Thanks for the encouragement and guidance about how to improve our study and manuscript.

Comments:

(1) LINE 257-258: If the control group and treatment group reached exponential phase at the same time? Has the growth curve made for this strain when cultivated with different concentration of NaCl? How long have you cultivated these groups and when did you determine their transcription?'

Response: Yes, thanks for your kindly reminding and helpful suggestions.

Solution: The cell growth curves with different salts were tested and the growth curves were achieved by detecting of absorbance at 600 nm as the following figure. And, for determine their transcription the cell was cultured in LB/LB5 (with 5% NaCl) medium collected when OD600 achieved to 0.6.

(2) LINE 258: The de color bar should be marked clearly some detailed information, such as 'relative expression', 'log2 fold' or 'xxx (brief information of treatment group) vs xxx (brief information of control group)'.

Response: Yes, thanks for your kindly reminding. The detail information has been added in lines 250.

Solution: ‘Log2 fold’ and the description in the line 250 were added as “fold change in Log2”.

(3) LINE 261: ‘catalvtic activity’ in Figure 1B should be ‘catalytic activity’?

Response: Sorry for the mistake and we revised it.

Solution: The ‘catalvtic activity’ in Figure 1B has been replaced by ‘catalytic activity’.

(4) LINE 262: Which seven branches? Do you mean cellular process, environmental information processing, genetic information processing, human diseases, metabolism and organismal systems? If yes, it should be six.

Response: Sorry for the mistake. There are six branches in the KEGG pathway classification.

Solution: The ‘seven’ has been revised as ‘six’.

(5) LINE 263: Do authors mean that genes from ‘global an overview maps’ occupied the largest proportion? The global maps (map numbers 01100s) and the overview maps (map numbers 01200s) are a special class of metabolic pathway maps in the KEGG PATHWAY database, presenting global and overall pictures of metabolism. They contain a large number of genes and enzymes from diverse pathways instead of a specific metabolism. So, what is the significance of this conclusion?

Response: Thank you for your careful reading and good advices.

Solution: This is a part of the transcriptional analysis. In order to reveal the main function of the genes, all the up-regulated genes were devised by ‘KEGG metabolic pathway’ analysis. The genes about ‘global and overview maps’, which containing ‘ABC transporters (ko02010) and two-component systems (ko02020)’, were the most abundant. Therefore, we proposed that the mechanism of the salt stress response for *Vibrio natriegens* were about transportation and moveability.

(6) LINE 265: third?

Response: Yes, thanks for your kindly reminding.

Solution: The 'third' in line 257 has been modified to 'first'.

(7) LINE 267: why did the authors use 4-fold as the cut-off?

Response: Thank you for your careful reading. The propose of the transcriptional analysis in this study was to find the promoters response to salt (NaCl). And, a part of genes, whose transcript level was fold changed below 4-fold by the growth state of the cell, but not salt response. So, '4-fold' was chosen as a cut-off line.

(8) LINE 267-273: Any references provided evidence to prove these 3 gene clusters were related to salt stress?

Response: Yes, thanks for your kindly reminding.

Solution: References were added as 6-10 to support the results in the second paragraph 2 of introduction.

(9) LINE 273-275: References are needed.

Response: Yes, thanks for your kindly reminding.

Solution: References were added as 6-10 to support the results in the second paragraph 2 of introduction.

(10) LINE 276: Vmax should be 'V. natriegens Vmax' or 'strain Vmax', similar mistakes in the manuscript should be corrected.

Response: Yes, thanks for your kindly reminding.

Solution: The writing format of 'Vmax' in the manuscript were unified to 'strain Vmax'.

(11) LINE 266-279: References are needed.

Response: Yes, thanks for your kindly reminding.

Solution: References were added as 6-10 to support the results in the second paragraph 2 of introduction.

(12) LINE 286: Did the authors mean that the direction of these two genes was opposite? Up-regulated opposite gene clusters?

Response: Yes, thanks for your kindly reminding.

Solution: As shown in Figure 2A, there are two up-regulated gene clusters in opposite directions.

(13) LINE 289: Names of genes or gene clusters should be marked clearly in Figure 2A.

Response: Yes, thanks for your kindly reminding, and we added the gene names in the revised manuscript.

Solution: The gene names were added in the Figure 2A as following:

(14) LINE 301: The unit of fluorescence intensity (y-axis) was missed.

Response: Yes, thanks for your kindly reminding.

Solution: The unit for fluorescence intensity was added as '(Arbitrary Unit)' in Figure 2C.

(15) LINE 302: Which P2 promoter? P2-2 or P2-1? In Figure 2C, which one was the promoter P2-1 or P2-2 used? And the legend of figure 2 should be more detailed. What did the P1 P2 P21 P12 mean in Figure 2C?

Response: Yes. In line 302 the ‘P2 promoters’ means the promoter P2-2.

Solution: In lines 290-294 and legend of Figure 2C, the details information about the promoters has been added as ‘Under 3% Na⁺ stress, the truncated regions, which contain only the promoters P1 (400 bp gene sequence front cluster *proWXV*), P2-1 (reverse 400 bp sequence front gene cluster *ectBACD*), P2-2 (reverse 400 bp sequence front cluster *proWXV*), or the full-length interval area (P12, and its reversed sequence was P21), have transcriptional activity (Figure 2C).’

(16) LINE 309: why not use the P2-2? Any result showed that P2-1 was better than P2-2?

Response: According to the previous studies, the CP and the HBCDs degradation were limited in the first steps (supported by Mpd or CYP168A1), separately. The transnational activity of P2-1 was more intense than that of P1, it can help to enhance efficiency of Mpd or CYP168A1 by expressing relatively more proteins.

(17) LINE 313: What did the ‘x+y=6’ mean in Figure 3C?

Response: Thanks for the helpful suggestions. In our previous study about the dehalogenating mechanism of HBCDs catalyzed by CYP168A1. The Br iron was replaced by ·OH one after another based on detection of the six dehalonated metabolites pentabromocyclododecanols (PBCDOHs), tetrabromocyclododecadiols (TBCDDOHs), bromide ion, and so on. In order to simplify the written form of the catalyze progress, the number of remained Br was corned as x, and the number OH- was corned as y.

(18) Figure 3D: 1.0

Response: Yes, thanks for your kindly reminding.

Solution: The '1' has been replaced by '1.0' in Figure 3D.

(19) LINE 320-322: The 'PET hydrolase' mentioned here is PETase? These products might be generated by PETase and MHETase. PETase cleaves BHET to MHET and EG, and the soluble MHET product is further hydrolyzed by MHETase to produce TPA and EG. And references are needed here.

Response: Yes, thanks for your kindly reminding. The PET hydrolases used in this study were introduced in section '**2.5 Construction of degradation models**', as 'The PET hydrolases, PETase²¹, MHETase²², LCC²³ and Tfca²⁴'.

Solution: To make it much clear for the function of the hydrolases, the 'As PETase or LCC could catalyze PET to mono-(2-hydroxyethyl) terephthalic acid (MHET) and bis-(2-hydroxyethyl) terephthalic acid (BHET) as main products, then the MHETase or Tfca can catalyze BHET and MHET further to ethylene glycol (EG), terephthalic acid (TPA), and other components under mild conditions (Figure 4A).' was added. And references were also added in lines 314-318.

(20) LINE 329: What are the differences between MPP and PPM? Exchanged the location of promoters? But the gene cluster of PPM and MPP shown in Figure 4B seems same.

Response: Yes, thanks for the good suggestions, the difference between PPM and MPP was the location of the promoters.

Solution: There is a mistake of gene cluster in Figure 4B, and it has been corrected as following.

(21) Figure 5B: missing the unit of the Y-axis. It seems that the error bars were missed.

Any repeats about this test?

Response: Yes, thanks for your kindly reminding. The Figure 5B was replaced by a corrected one.

Solution: MTT method was used commonly to detect biomass as in reference 32. AS ‘The biomass binding on chitin was measured via a (3-(4,5)-dimethylthiazol-2-yl)-2,5-diphenyltetrazoliumbromide (MTT) assay kit. The strain was incubated and added with 100 μ L MTT (5 g/L) and kept in the dark for 4 ~ 5 h, following that 100 μ L DMSO was added and read at 570 nm. The mortality percentage was calculated³².’

(22) LINE 566: ‘Environ Sci Technol’ should be italic as above.

Response: Yes, thanks for your suggestion.

Solution: The ‘Environ. Sci. Technol.’ in line 570 was corrected to ‘*Environ. Sci. Technol.*’

(23) LINE 568-569: The format of this reference should be unified.

Response: Yes, corrected.

Solution: The format of reference in lines 568-569 has been replaced by ‘Wang, J., Tan, Z., Peng, J., et al. The behaviors of microplastics in the marine environment. *Mar. Environ. Res.* **113**: 7–17 (2016).’

(24) Figure 5A: The information of the sample in each lane should be detailedly described in the legend.

Response: Yes, thanks for your kindly reminding. Detail information was added in Fig. 5.

Solution: The figure legend 5 has been replaced by ‘(A) Analysis of the expression of the chitin-binding protein GbpA by SDS-PAGE. M, protein marker; lane 1-3, crude enzyme, supernatant, precipitate of strain Vmax; lane 4-6, crude enzyme, supernatant, precipitate of strain Vmax-GbpA.’

(25) Figure 5C: The scale was missed.

Response: Yes, sorry for the mistake. The missed scale was added.

Solution: Figure 5C was revised as following:

Response to Reviewer 2

Bioremediation of marine pollutants like microplastics, pesticides, or halogenated additives like commonly used brominated flame retardants under high salt conditions is an exciting, relevant subject which would be of high interest to the readership of Communications Biology and comply entirely with the journal profile. Even though many articles have been published concerning the topics, especially compiling aspects of new discovered high salt-induced functional promoters, the rapidly growing marine bacterium *Vibrio natriegens* strain Vmax with high potential as a biotechnological production platform, highly relevant pollutant degradation, and the sustainable strategy of biomass recycling of the biocatalyst could make a significant contribution. The manuscript is written in an engaging and lively style. Still, it would benefit from some persuasive proofreading, recent comprehensive references and detailed data analysis and interpretation, and more explicit, compelling arguments in the results and discussion parts as parts of the manuscript made it challenging to follow.

My recommendation to the editor is to reject this paper. In my opinion, the essential revisions are too fundamental for the submission to continue being considered in its current form. As a research article, the manuscript should provide an overview of the published literature, significant advances in the genetics of halophilic microbes, biodegradation of microplastics, other xenobiotics (e.g. pesticides, flame retardants), their combined application and presentation in technically sound data. The latter is missing in many aspects of the current paper, but especially in the last two chapters of the Results section: Salt-induced degradation models for PET and the Environmental and ecological safety of engineered bacteria. Many aspects of the results are poorly discussed or completely ignored, like the biomass immobilization on chitin, the biocatalyst recycling and the title announced feature of controllable and environmental safe application of GMOs is missing.

Additionally, as a reader and reviewer, the current paper lacks originality (PET-biodegradation in marine environments has been a focus of many recently published papers; same for HBCD and chlorpyrifos), the presentation needs significant improvements, and the potential impact is low.

I would strongly advise the authors to re-write their introduction, discussion and conclusions. The chapters concerning halophilic responses in *V. natriegens*, salt-induced promoters, and biodegradation are inspiring and should be enhanced by providing more details. The data regarding mPET-degradation are attractive as well. Still, the lack of consistency, especially the density of data points in the first period of 0h–72h, is unfortunate. These experiments should be reiterated for the best of this manuscript.

Response: Thanks for the encouragement and guidance about how to improve our study and manuscript. The introduction, discussion and conclusions were re-written according to the new references and results. And, the PET degrading experiments with whole cell and crude enzyme systems were re-conducted to supple data of the accumulation of products in 0-72 h to make it complete, the results were shown in the new Figure 4B and 4C.

Comments:

(1) same time frame for comparison => 24h.

Response: Yes, thanks for your kindly reminding.

Solution: The ‘1 mg/L HBCDs in 4 h’ was matched to the degrading curve of *Vmaxcyp168A1p12FdFNR* in Figure 3D, the 1mg/L HBCDs were fully degraded in 4 hours.

(2) Is that so for sure?, rewrite and rethink whole sentence/conclusion

Response: Yes, thanks for your good suggestions.

Solution: ‘This study helps answer the problem of low growth and improves the degradation rates of natural microorganisms, suggesting its application potential in bioremediation of marine pollution.’ has been replaced by ‘This study may help answer the usage of rapidly growing marine bacteria such as *V. natriegens* Vmax to degrade marine pollutions efficiently.’

(3) different more recent reference needed!

Response: Yes, thanks for your kindly reminding and we newly added the reference.

Solution: The reference 1 was replaced by a recent reference as ‘Oliveira, J., Belchior, A., Silva, V., et al. Marine environmental plastic pollution: mitigation by microorganism degradation and recycling valorization. *Front. Mar. Sci.* **7**: 567126 (2020).’

(4) different more recent reference needed!

Response: Yes, thanks for your kindly reminding.

Solution: The sentence cited reference 2 has been deleted.

(5) Please remove UNEP 2018 and use figures from the recent UNEP report concerning the projections for the plastic waste annually released to oceans tripled by 2040 or at least doubled by 2030 starting with 9-14MT in 2016 and cite correctly.

Response: Yes, thanks for your kindly reminding.

Solution: This paragraph has been removed and replaced by a new as ‘Marine pollution, including heavy metals, petroleum, persistent organic pollutants (POPs), debris, and radionuclides, can directly or indirectly be harmful to live organisms and resources². Plastic pollution has escalated during last 50 years, and estimated contents of plastic in marine are more than 250,000 tons³. Removal of microplastics using sorption and filtration has been constructed, such as absorbing the microplastic particles on the surface of marine green algae, filtration of microplastics by membrane technology, and even being combined with membrane bioreactors, with the remove efficiency reaching to 97.2%⁴. Two types of *Bacillus* strains isolated from the mangrove sediments were found to degrade different microplastics with the reduction only at 0.0019 mg/ day⁵. However, the degradation of microplastics in marine water has few studied, and only a limited number of bacteria are able to degrade the contaminants under marine conditions with a salinity ranges between 3.3–3.7%.’ The biodegrading of microplastics has been reviewed in this section.

(6) Please argue more detailed, do you plan to inoculate marine litter or do you mean

that only a limited number of bacteria is able to degrade the contaminants under marine conditions with an salinity ranges between 3.3 - 3.7%? Rewrite sentence.

Response: Yes, thanks for your kindly reminding. The sentence has been advised.

Solution: ‘Owing to the high salt content of marine water, only a limited number of bacteria can be used to degrade the contaminants.’ was replaced by ‘However, the degradation of microplastics in marine water has few studied and only a limited number of bacteria is able to degrade the contaminants under marine conditions with an salinity ranges between 3.3–3.7%’.

(7) what is this reference 9 for?

Response: Yes, thanks for your kindly reminding.

Solution: The reference 9 in section 2.1 was deleted.

(8) Correct?

Response: Yes, thanks for your kindly reminding. There is a mistake has been advised about the consistence of NSS.

Solution: The sentence ‘The 0.5 × nigh salts solution medium (NSS) contained (per liter) 8.8 g NaCl, 0.735 g Na₂SO₄, 0.125 g KCl, 0.02 g KBr, 0.935 g MgCl₂·6H₂O, 0.205 g CaCl₂·2H₂O, 0.004 g SrCl₂·6H₂O, and 0.004 g H₃BO₃’ has been replaced by ‘The nigh salts solution medium (NSS) contained (per liter) 8.8 g NaCl, 0.735 g Na₂SO₄, 0.125 g KCl, 0.02 g KBr, 0.935 g MgCl₂·6H₂O, 0.205 g CaCl₂·2H₂O, 0.004 g SrCl₂·6H₂O, and 0.004 g H₃BO₃’.

(9) What do you mean with 1L cells (cell numbers/ml or OD??)

Response: Yes, detail information has been added.

Solution: The sentence ‘All the samples were cultured in 30°C thermostatic shakers at 200 rpm. Cells were harvested when OD₆₀₀ reached to 0.6 ~ 0.8 for transcriptome sequencing, frozen by liquid nitrogen and stored at -80°C before sequencing.’

(10) please rewrite in more detail how you prepared your cells, temperature,

quenching, RNA-isolation; How did you prevent oxygen limitation if you cultivate that way? what was the shaking rpm...? important details for reproducibility!

Response: Yes, thanks for your kindly reminding.

Solution: The detail information has been added, as ‘All the samples were cultured in 30°C thermostatic shakers at 200 rpm’.

(11) What was the cut-off for fold-changes? Link to suppl. Material missing.

Response: Thank you for this thought-provoking advice.

Solution: A sentence ‘Genes with high transcription levels under high salt concentration (5% NaCl) (up-regulation fold change ≥ 4) were summary as the research candidates.’ was added.

(12) =

Response: Yes, thanks for your kindly reminding.

Solution: The ‘LB5’ was replaced by ‘5% NaCl lysogeny broth (LB5)’.

(13) thus degradation assay took place in 0.9% NaCl?

Response: Yes, thanks for your kindly reminding.

Solution: The degrading rates were detected in the 0.5 × NSS buffer.

(14) ectoine.

Response: Yes, thanks for your kindly reminding.

Solution: The ‘ecotone’ has corrected to ‘ectoine’.

(15) Consider a different label, not clear

Response: Yes, thanks for your kindly reminding.

Solution: The tittle 3.3 has been revised to ‘Salt-induced degradation models for CP and HBCDs with promoters P1 and P2-1’.

(16) converted?

Response: Yes,

Solution: The 'catalyzed' in line 308 has been revised to 'converted'.

(17) Consider a different label, not clear

Response: Yes, thanks for your kindly reminding.

Solution: The title 3.4 'Salt-induced degradation models for PET in version Vx2.0' has been replaced by 'Salt-induced degradation models for PET with promoters P1 and P2-2'.

(18) Is considering the low sampling rate this ranking is not proven; please repeat and provide more insight by sampling the first 4 days or remove, that way no rate can be determined; how do you prove that BHET/MHET concentrations higher between and decreased later as shown in cell crude extracts 0-4d? correct Fig. 4B labels x/y axis

Response: Yes, thanks for your kindly reminding.

Solution: The degradation rates of the four engineered strains were re-detected using whole cell system, the results were shown in the new Figure 4B. And, the degrading rates were re-compared and ranked, ' V_{\max} -MHETaseP122PETase (MPP) > V_{\max} -PETaseP122MHETase (PPM) > V_{\max} -TfcaP122LCC (TPL) > V_{\max} -LCCP122Tfca (LPT)'.

(19) Here group: cells/constructs

Response: Yes, thanks for your kindly reminding.

Solution: The 'group' has been replaced by 'constructions'.

(20) PPM group? Find a different name

Response: Yes, changed.

Solution: All the PPM group in the manuscript have been replaced by 'PPM constructions'.

(21) PPM activity in 4C looks like an artefact based on one measurement/sampling

point in 48h; Please repeat analysis!

Response: Yes, thanks for your kindly reminding.

Solution: The sentence ‘At this point, degradation ceased’ was deleted.

(22) Comparing crude extract activities:....

Response: Yes, corrected.

Solution: The sentence has been advised to ‘The activities for crude enzymes were stronger than activities of the whole cells, owing to the suitable temperature for enzymes. Changes in the surface morphology of PET membrane samples, treated by strains PPM, MPP, LPT, and TPL, were shown in Figure 4D. Compared to untreated PET, varying degrees of fragmentation were detected for the treated samples. For the PET membrane degradation, the most obvious change was observed in the MPP samples, which matched to the accumulation of degrading products.’

(23) detected

Response: Yes, thanks for your kindly reminding.

Solution: Actually, nano-PET powder and PET membrane were coexisting in any single PET degrading system, the degrading rates were compared according to detection of total products. In order to make it clearer. ‘The most obvious change was observed in the MPP samples. The degradation rates of the four engineered strains ranked the same as with the whole cell degradation systems.’ was changed to ‘For the PET membrane degradation, the most obvious change was observed in the MPP samples.’

(24) But you ranked MPP 2nd; looking at the images PPM looks untouched compared to PET; LPT and TPL show very low degrees of fragmentation

Response: Yes, thanks for your kindly reminding.

Solution: Actually, nano-PET powder and PET membrane were coexisting in any single PET degrading system, the degrading rates were compared according to detection of total products. In order to make it clearer ‘The most obvious change was

observed in the MPP samples. The degradation rates of the four engineered strains ranked the same as with the whole cell degradation systems.’ was changed to ‘For the PET membrane degradation, the most obvious change was observed in the MPP samples.’

(25) how do you determine this? please provide details

Response: Yes, thanks for your kindly reminding.

Solution: As mentioned in section ‘2.6 Immobilization and continuous recycling’, The biomass binding on chitin was measured via a (3-(4,5)-dimethylthiazol-2-yl)-2,5-diphenyltetrazoliumbromide (MTT) assay kit. The strain was incubated and added with 100 μ L MTT (5 g/L) and kept in the dark for 4 ~ 5 h, following that 100 μ L DMSO was added and read at 570 nm. The mortality percentage was calculated³².

(26) How did you analyse this?

Response: Yes. As mentioned in section ‘2.6 Immobilization and continuous recycling’, ‘the suspension was taken as a sample for detecting the corresponding substrates.’ To make it clear, a sentence ‘To test the efficiency of recycled bacteria, the engendered HBCDs degrading strain was taken as module to calculate the HBCDs remaining rates for three times, by detecting the remained HBCDs concentrations.’

(27) Chromatograms in 5F are not labeled correctly; ranking can not be verified, if TPL is on bottom why not ranked higher? Please change!

Response: Yes, thanks for your kindly reminding.

Solution: Here, in Figure 5F, the chromatograms were drawn to prove that the recycled bacteria has corresponding activities, without X axis was added. The compaction of the degrading rates was not a purpose for this test, so the last sentence was deleted.

(28) rewrite completely; split in shorter sentences

Response: Yes. Lines 423-431, has been corrected.

Solution: The sentence has been replaced by ‘Most of the reported salt tolerance bacteria could secrete compatible small molecules to adapt to the salt stress, such as *Methylovimicrobium alcaliphilum* 20Z (NC_016112.1)³⁵, *Bacillus halodurans* C-125 (BA000004.3)³⁶, *Streptomyces coelicolor* A3 (AL645882.2)³⁷, *Halomonas elongata* HEK1 (FN869568.2)³⁸, and *Chromohalobacterium salexigens* DSM 3043 (CP000285.1)³⁹. Ectoine biosynthesis genes (*ectABC* cluster), located in the genome of the above bacteria, are transcribed under the control of the salt-induced promoters adjacent to the “ATG” codon of *ectA*, such as *ectApI*, *PectA*, *PectA*, *PectAI-4* and *PectB*, and *PectAI-4* and *PectB*.’

(29) Please discuss in more detail referring to Fig. S1!

Response: Yes, more discuss in detail referring to Fig.S1 has been added.

Solution: ‘The location of genes related to salt stress in strain Vmax presents an obviously location-specific advantage, compared to the other reported halophilic or

halotolerant microorganisms. The genes related to Na⁺/K⁺ transcription and ectoine, proline, and betaine biosynthesis could be induced to express nearly at the same time.’ has been added.

(30) rename

Response: Yes.

Solution: The ‘degrading model II’ has been replaced by ‘HBCDs degrading model’.

(31) Please discuss safety issues of the release of genetically engineered microbes to the environment!

Response: Yes, a sentence was added to illustrate and discuss the safety issues of the engineered bacteria.

Solution: Safety issues of the release of genetically engineered microbes was discussed as ‘The usage of gene-edited bacteria in the environment may suffer some biosecurity risks. They may refer to unauthorized access, loss, theft, misuse, diversion or intentional release. If a biosecurity accident happens, it would pose a huge threat to humans and nature⁴¹.’ And a reference was cited as reference 41.

(32) Please provide a recent reference

Response: Yes, the sentence has been deleted.

(33) this statement is incorrect as there are many papers out with an focus on applicability of PET-hydrolases; please rewrite and cite recent publications! there might be a lack on applications in high osmotic settings but this needs more arguments espec. as some PET-hydrolases hosting bacteria are of marine origin and there is a lot of research done on biofilms and marine litter, garbage patches (e.g. Great Pacific/North-Atlantic...)

Response: Yes, it has been corrected.

Solution: The sentence has been replaced by ‘Although many kinds of PET hydrolases have been investigated in various microorganisms, including cutinase,

lipase, and PETase. The applications of those enzymes to engineering bacteria degrading PET in the marine environment are limited.'

(34) Please discuss the implications of applying these strains in more detail espec. the recycling of biomass chitin bound for bioremediation needs deeper discussion and comparison with alternative methods.

Response: Yes, the advantages of immobilization engineered cells on materials was added.

Solution: 'however, the immobilized cells may exhibit improved catalytic activities, shorten bioremediation time, lower production cost with lower biosecurity risks⁴³.' has been added.

(35) Legend not selfdecriotive w/o text; provide more details 1D) V. natriegens Vmax is gram-negative gamma proteo-bacterium of the Vibrionales order, what do you mean with nucleus; as most eucaryotic DNA-folding related factors are absent in bacteria it is more correct to speak of nucleus-like or nucleoid structure, because it is mainly organized and maintained by nucleoid-associated DNA-binding proteins, supercoiling and transcription processes, but no nucleus (Please correct!)

Response: Yes, thanks for your kindly reminding.

Solution: The 'nucleus' in Figure 1D has been replaced by 'genomic DNA'. And the figure legend for Figure 1 has been revised to 'Transcriptome data analysis. (A) The heat map, based on hierarchical analysis using the \log_2 (5%/1%) for each gene at two groups (5% and 1%), depicted that 1,596 genes were detected out. The colors were ranging from blue to red, representing the values of \log_2 (5%/1%). (B) Significantly enriched GO categories for the genes with fold changes in $\text{Log}_2 \geq 4$. (C) Differential gene KEGG pathway classification for the genes with fold changes in $\text{Log}_2 \geq 4$. (D) Proposed halophilic mechanism of strain Vmax based on the transcriptome analysis, including selectively transporting the inorganic ions Na^+/K^+ in the cytoplasm to the accumulation of specific organic substances of low-molecular weight, such as ectoine, proline, betaine, trehalose, choline-O-sulfuric acid, and carnitine. Information

regarding the transcriptional regulation of biosynthesis is overwhelmingly about the biosynthesis of ectoine’.

(36) Pease correct: HBCDs degradation in graph

Response: Yes, thanks for your kindly reminding.

Solution: The ‘HBCDs degrading’ in figure 5D has been replaced by ‘HBCDs degradation’.

(37) How do you calculate a rate and what is its dimension; what did you analyse Br--release or degradation products; 5F descriptions MPP, LPT and TPL missing, please correct.

Response: Yes, thanks for your kindly reminding.

Solution: As in the answers 25 and 26, ‘The biomass binding on chitin was measured via a (3-(4,5)-dimethylthiazol-2-yl)-2,5-diphenyltetrazoliumbromide (MTT) assay kit. The strain was incubated and added with 100 μ L MTT (5 g/L) and kept in the dark for 4 ~ 5 h, following that 100 μ L DMSO was added and read at 570 nm. The mortality percentage was calculated³².’ and ‘To test the efficiency of recycled bacteria, the engendered HBCDs degrading strain was taken as module to calculate the HBCDs remaining rates for three times, by detecting the remained HBCDs concentrations.’ And, descriptions for MPP, LPT and TPL were added in Figure 5F.

(38) 4B) Please provide graphs in same y-axis dimensions for comparison; do not use animated graphs; please repeat measurements espec. for TPL only one measurement? sample rate 0-5 days insufficient, higher density needed!

Response: Yes, thanks for your kindly reminding.

Solution: The experiment has been repeated, and the results has been shown in the new Figure 4B.

Reviewers' comments:

Reviewer #1 (Remarks to the Author):

Through author's efforts, most of my concerns have been addressed. Some flaws mentioned before should still be resolved.

LINE 46: mg/day

LINE 61: What was inserted into the large chromosome of *V. natriegens*. Please rewrite and make this sentence clear.

LINE 255: Gene in 'global and overview maps' contained too many genes in life (<https://www.genome.jp/pathway/map01110>), therefore 'of which the genes related to global and overview maps occupied the largest proportion' might not mean anything. Delete this sentence and just describe ABC transporters and two-component systems would be better.

LINE 259: We have discussed this concern last time. The authors' response is 'The propose of the transcriptional

analysis in this study was to find the promoters response to salt (NaCl). And, a part of genes, whose transcript level was fold changed below 4-fold by the growth state of the cell, but not salt response. So, '4-fold' was chosen as a cut-off.' Any specific table or figure present these statistical data?

LINE 259: Please cite related references here. Ref 6-10 just used to support 'Halophilic or halotolerant bacteria are capable of accumulating high concentrations of various organic osmotic solutes (OOSs)'. But here, another viewpoint was showed, specific references should be cited again.

LINE 308. HBCDs could be converted to ... or 1 mg/L HBCDs could be completely converted to...

Reviewer #2 (Remarks to the Author):

As stated in my first review of this manuscript, the authors tackle the fascinating topic of marine biodegradation to solve marine pollution of microplastics, pesticides, or halogenated additives like commonly used brominated flame retardants. They discovered and described functional promoters under increased salt stress. Subsequently, they used them in different constructs of the fast-growing marine bacterium *Vibrio natriegens* strain Vmax in biodegradation approaches. Finally, the authors developed a system to recycle the engineered strains by an immobilization strategy.

After the major revision, many flaws have been fixed, large parts of the introduction and discussion part have been rewritten, more recent references added, and main experiments repeated, all for the benefit of the revised manuscript.

I want to thank the authors for their strong efforts, as significant manuscript improvements can be stated. From today's perspective, especially comparing the results of the repeated experiments depicted in Fig. 4 with the ones of the original paper are much more significant now and make the manuscript stronger. Different PET-hydrolases combined with salt-induced promoters in cell-based (4B) and cell-free systems (4C) achieved much higher (>10x) concentrations of degradation products (plastic monomers) in the repeated experiments.

However, some minor flaws, mainly typos, wrong wordings, phrases etc. are still present and need to be fixed, and the revised manuscript would benefit from some persuasive proofreading.

L23, 367: high salt concentrations (> 1%): after % w/v missing

L39: wrong wording: please change: live -> living organisms

L44-48: wrong wording: please change: remove -> removal; delete to%; add: has few been studied

L54: wrong wording: please change: delete resulting in

L55-56: wrong description: slight halophiles is the wrong scientific terminology, please change to: halotolerant

L61: wrong phrase, please change: the large chromosome of *V. natriegens* is incorrect as the strain has only one chromosome, please change large chromosome to genome

L101-102: Please change phrase Luria-Bertani to the correct: lysogeny broth (as was done in rest Manuscript e.g., L151

L108: Please change: summary -> summarized

L149, 162, 184, 325-334 Please consider changing PPM, MHET, TPL, BHET-constructions to PPM, MHET, TPL, BHET-constructs

L163: Please remove one set of 44°C, 55°C: double written

L193: Wrong wording, please change: remained HBCDs-concentration to remaining HBCDs...

L309: Please correct typo: dibrominated to debrominated

L379: Change salt tolerance bacteria to salt tolerant bacteria

L405: Please correct: bacteria to the singular bacterium

L435: Please correct to : are limited. However,

Response to Reviewer 1

Through author's efforts, most of my concerns have been addressed. Some flaws mentioned before should still be resolved.

Response: Thanks for the encouragement and guidance about how to improve our study and manuscript.

Comments:

(1) LINE 46: mg/day.

Response: Yes, thanks for your kindly reminding and helpful suggestions.

Solution: the 'mg/ day' has been changed to 'mg/day'.

(2) LINE 61: What was inserted into the large chromosome of *V. natriegens*. Please rewrite and make this sentence clear.

Response: Yes, thanks for your kindly reminding. The sentence has been revised to make it clear.

Solution: 'When cassettes contain a T7 RNA polymerase gene, under the control of isopropyl-beta-D-thiogalactopyranoside (IPTG) or arabinose inducible promoters (lacUV5 and araBAD), were inserted into the large chromosome of *V. natriegens* (ATCC 14048, the original strain of *V. natriegens* Vmax), robust GFP (introduced in the pET vector) expression was detected. As a result, the engineered strain Vmax can be used as a rapidly platform for genetic engineering¹².' has been replaced by 'Cassettes containing T7 RNA polymerase gene under the control of either an IPTG- or arabinose-inducible promoter (lacUV5 and araBAD, respectively) were inserted into the large chromosome of *V. natriegens* (ATCC 14048, the original strain of *V. natriegens* Vmax). When expression plasmids containing GFP gene under the control of T7 promoter was introduced, robust GFP expression was detected¹².'

(3) LINE 255: Gene in 'global and overview maps' contained too many genes in life (<https://www.genome.jp/pathway/map01110>), therefore 'of which the genes related to

global and overview maps occupied the largest proportion' might not mean anything. Delete this sentence and just describe ABC transporters and two-component systems would be better.?

Response: Yes, thanks for your kindly reminding.

Solution: The 'of which the genes related to global and overview maps occupied the largest proportion' was deleted.

(4) LINE 259: We have discussed this concern last time. The authors' response is 'The propose of the transcriptional analysis in this study was to find the promoters response to salt (NaCl). And, a part of genes, whose transcript level was fold changed below 4-fold by the growth state of the cell, but not salt response. So, '4-fold' was chosen as a cut-off.' Any specific table or figure present these statistical data?

Response: Sorry for the last answer was not explained the '4-fold' clear for the question: why did the authors use 4-fold as the cut-off ?

Solution: The propose of the transcriptional analysis in this study was to find the promoters response to salt (NaCl), so the strain Vmax was incubated in media with 1 and 5% (w/v) final concentration of NaCl, separately. The transcriptional results containing a part of genes, whose transcript level was fold changes in $\text{Log}_2^{5\%/1\%}$ below 4. Combined the gene function annotation and test error, the genes mention above were considered has no relation to salt response. So, '4-fold' was chosen as a cut-off line.'

(5) LINE 259: Please cite related references here. Ref 6-10 just used to support 'Halophilic or halotolerant bacteria are capable of accumulating high concentrations of various organic osmotic solutes (OOSs)'. But here, another viewpoint was showed, specific references should be cited again.

Response: Thank you for your careful reading and good advices.

Solution: Reference was added as 34 in line 259.

(6) LINE 308. HBCDs could be converted to ... or 1 mg/L HBCDs could be

completely converted to...

Response: Yes, thanks for your kindly reminding.

Solution: The sentence has been replaced by ‘ HBCDs (1 mg/L) could be completely converted to debrominated products, including pentabromocyclododecanols (PBCDOHs) and tetrabromocyclododecadiols (TBCDDOHs) within 8 h (Figure 3D)’.

Response to Reviewer 2

As stated in my first review of this manuscript, the authors tackle the fascinating topic of marine biodegradation to solve marine pollution of microplastics, pesticides, or halogenated additives like commonly used brominated flame retardants. They discovered and described functional promoters under increased salt stress. Subsequently, they used them in different constructs of the fast-growing marine bacterium Vibrio natriegens strain Vmax in biodegradation approaches. Finally, the authors developed a system to recycle the engineered strains by an immobilization strategy.

After the major revision, many flaws have been fixed, large parts of the introduction and discussion part have been rewritten, more recent references added, and main experiments repeated, all for the benefit of the revised manuscript.

I want to thank the authors for their strong efforts, as significant manuscript improvements can be stated. From today’ s perspective, especially comparing the results of the repeated experiments depicted in Fig. 4 with the ones of the original paper are much more significant now and make the manuscript stronger. Different PET-hydrolases combined with salt-induced promoters in cell-based (4B) and cell-free systems (4C) achieved much higher (>10x) concentrations of degradation products (plastic monomers) in the repeated experiments.

However, some minor flaws, mainly typos, wrong wordings, phrases etc. are still present and need to be fixed, and the revised manuscript would benefit from some persuasive proofreading.

Response: Thanks for the encouragement and guidance about how to improve our study and manuscript.

Comments:

(1) L23, 367: high salt concentrations (> 1%): after % w/v missing.

Response: Yes, thanks for your kindly reminding.

Solution: The 'w/v' was added behind 1% in lines 23 and 367.

(2) L39: wrong wording: please change: live -> living organisms.

Response: Yes, thanks for your good suggestions.

Solution: 'live' in line 39 has been revised to 'living'.

(3) L44-48: wrong wording: please change: remove -> removal; delete to ...%; add: has few been studied.

Response: Yes, thanks for your kindly reminding.

Solution: The 'remove' in line 45 has been replaced by 'removal', the 'to' in line 45 has been deleted, and 'been' was added in line 48.

(4) L54: wrong wording: please change: delete resulting in.

Response: Yes, thanks for your kindly reminding.

Solution: The 'resulting in' in line 54 was deleted.

(5) PL55-56: wrong description: slight halophiles is the wrong scientific terminology, please change to: halotolerant.

Response: Yes, thanks for your kindly reminding.

Solution: The 'Halophilic or halotolerant' in line 55 was replaced by 'Halotolerant'.

(6) L61: wrong phrase, please change: the large chromosome of *V. natriegens* is incorrect as the strain has only one chromosome, please change large chromosome to genome.

Response: Yes, thanks for your kindly reminding. While, as reported, there are two chromosomes with different size, in *V. natriegens*. Here, the cassettes containing the

gene encoding T7 RNA polymerase was inserted in the large chromosome.

(7) L101-102: Please change phrase Luria-Bertani to the correct: lysogeny broth (as was done in rest Manuscript e.g., L151).

Response: Yes, thanks for your kindly reminding.

Solution: The 'Luria-Bertani' in line 101 has been replaced by 'lysogeny'.

(8) L108: Please change: summary -> summarized.

Response: Yes, thanks for your kindly reminding.

Solution: The 'summary' in line 108 has been replaced by 'summarized'.

(9) WL149, 162, 184, 325-334 Please consider changing PPM, MHET, TPL, BHET-constructions to PPM, MHET, TPL, BHET-constructs.

Response: Yes, detail information has been added.

Solution: All the 'constructions' have been replaced by 'constructs'.

(10) L163: Please remove one set of 44°C, 55°C: double written.

Response: Yes, thanks for your kindly reminding.

Solution: The repeated '44°C, 55°C' was deleted.

(11) L193: Wrong wording, please change: remained HBCDs-concentration to remaining HBCDs...

Response: Thank you for this thought-provoking advice.

Solution: The 'remained' in line 193 has been replaced by 'remaining'.

(12) L309: Please correct typo: dibrominated to debrominated.

Response: Yes, thanks for your kindly reminding.

Solution: The 'dibrominated' in line 309 has been replaced by 'debrominated'.

(13) L379: Change salt tolerance bacteria to salt tolerant bacteria.

Response: Yes, thanks for your kindly reminding.

Solution: The 'tolerance' in line 379 has been replaced by 'tolerant'.

(14) L405: Please correct: bacteria to the singular bacterium.

Response: Yes, thanks for your kindly reminding.

Solution: The 'bacteria' in line 405 has corrected to 'bacterium'.

(15) L435: Please correct to : are limited. However,

Response: Yes, thanks for your kindly reminding.

Solution: The 'are limited, however,' in line 435 has corrected to 'are limited. However,'.